# Ethics as Lived Practice. Anticipatory Capacity and Ethical Decision-Making in Forensic Genetics

**DOI:** 10.3390/genes12121868

**Published:** 2021-11-24

**Authors:** Matthias Wienroth, Rafaela Granja, Veronika Lipphardt, Emmanuel Nsiah Amoako, Carole McCartney

**Affiliations:** 1Centre for Crime and Policing, Department of Social Sciences, Northumbria University, Newcastle upon Tyne NE1 8ST, UK; 2Communication and Society Research Centre, University of Minho, 4710-057 Braga, Portugal; r.granja@ics.uminho.pt; 3University College Freiburg, Albert-Ludwigs-Universität, 79098 Freiburg, Germany; veronika.lipphardt@ucf.uni-freiburg.de; 4Department of Applied Sciences, University of the West of England, Bristol BS16 1QY, UK; emmanuel.nsiahamoako@uwe.ac.uk; 5Science & Justice Research Interest Group, Law School, Northumbria University, Newcastle upon Tyne NE1 8ST, UK; carole.mccartney@northumbria.ac.uk

**Keywords:** ethics, forensic genetics, ethics as lived practice, decision-making, genetic databasing, forensic DNA phenotyping, forensic genealogy, forensic epigenetics, communication, database

## Abstract

Greater scrutiny and demands for innovation and increased productivity place pressures on scientists. Forensic genetics is advancing at a rapid pace but can only do so responsibly, usefully, and acceptably within ethical and legal boundaries. We argue that such boundaries require that forensic scientists embrace ‘ethics as lived practice’. As a starting point, we critically discuss ‘thin’ ethics in forensic genetics, which lead to a myopic focus on procedures, and to seeing ‘privacy’ as the sole ethical concern and technology as a mere tool. To overcome ‘thin’ ethics in forensic genetics, we instead propose understanding ethics as an intrinsic part of the lived practice of a scientist. Therefore, we explore, within the context of three case studies of emerging forensic genetics technologies, ethical aspects of decision-making in forensic genetics research and in technology use. We discuss the creation, curation, and use of databases, and the need to engage with societal and policing contexts of forensic practice. We argue that open communication is a vital ethical aspect. Adoption of ‘ethics as lived practice’ supports the development of anticipatory capacity—empowering scientists to understand, and act within ethical and legal boundaries, incorporating the operational and societal impacts of their daily decisions, and making visible ethical decision making in scientific practice.

## 1. Introduction

Ethical principles and their application in practice have come to matter to many individual forensic geneticists, working to ensure that, while the scientific foundation of the techniques they are developing and deploying for casework is reliable, they are also contributing to a just society. Yet forensic genetics as a discipline and a community of practice—research scientists, laboratory practitioners, journals, industry, professional bodies, and users of forensic genetic analyses in policing and elsewhere—continue to face scrutinize new and emerging forensic genetic technologies for their social, ethical, and legal ramifications, alongside their scientific and technical dimensions. For example, the German Government led an inquiry into forensic DNA phenotyping and biogeographic ancestry testing between 2017 and 2019, leading to the legalisation of the former in 2019, but not of the latter, due to concerns around racial profiling [1]. The Law Commission of New Zealand reported on new and emergent forensic genetics in 2020, recommending the comprehensive revision of legislation; increased oversight; and greater consideration of concerns surrounding privacy and discrimination [2]. The Law Commission of Ontario reviewed Artificial Intelligence (AI) and probabilistic genotyping DNA tools in June 2021, suggesting that inherent biases in these technologies challenge the notion of DNA profiling as a “gold standard” and requires that there is continuous scrutiny as well as new legislation [3]. The Scottish Government and the UK House of Lords have commenced parallel enquiries into developments in technologies for policing in 2021 [4,5], scoping opportunities, and challenges arising for policing from new technologies.

While these enquiries are undertaken, reports published, and debates held, demand continues for the utility of databases (including genetic ones) to be maximised, and the promise of forensic genetics—often portrayed as the greatest crime-fighting tool since fingerprinting—be realised and capitalised upon. Meanwhile, new techniques are being developed and debated. Despite remaining contentious, internationally, the use of genetic ‘phenotyping’ in police investigations is growing. The Swiss Parliament is the latest to prepare legislation to allow for forensic DNA phenotyping amidst, both, supportive and critical debate [6]. Similarly, since 2018, forays into forensic genetic genealogy have been spreading since the successful detection of the perpetrator with the help of the technique in the ‘Golden State Killer’ case in the USA [7,8,9,10].

In these areas, as well as generally, forensic scientists face institutional pressures, both from casework and their scientific community (including employers/funders), which must be met with scientific professionalism and ethical practice. Yet it was not long ago that scientists themselves conceded that forensic science ‘is not sufficiently well developed as a profession’ (p. 523, [11]). Within the scientific community, there is increased scrutiny regarding the way that data are sourced and shared, informing both forensic research and casework analysis [12]. The intensification of external scrutiny, and productivity pressures demand the highest levels of professionalism, which must include a strong, internalized code of ethics that has not yet been proven to be universal [13].

In some parts of the world, forensic genetics research and casework have become an important aspect of policing, criminal justice, and victim identification. In multiple ways, forensic genetics contributes to rendering these domains more agile vis-à-vis existing and emerging demands and challenges. Novel technologies are developed in response to scientific insights and practical problems while remaining subject to the same responsibilities that scientists and users have towards their specific institutions, disciplines, and to society as a whole.

Therefore, in this paper, we offer an approach through which the forensic genetics community can reflect on their daily decisions in both scientific research and casework as part of their professional and ethical forensic practice. While we cannot, nor do we aim to, provide technical or legal solutions, we offer a framework for ethical conduct. We consider the problem of ‘thin’ ethics in forensic genetics and propose understanding ethics as an intrinsic part of the lived practice of a scientist. We aim to support scientists in developing anticipatory capacity [14]—to understand the wider operational and societal impacts of their daily decisions around forensic genetics—by making visible the ethical decisions in both their scientific practice and in the application and implementation of forensic technologies. We do so by exploring ethical aspects of decision-making in forensic genetics and illustrating the discussion by way of three case studies of emerging forensic genetics technologies.

## 2. Materials and Methods

This paper builds on literature analysed in three previous reviews of social and ethical aspects of scientific, technological, and operational developments in forensic genetics [15,16,17], adding consideration of scholarship published since 2017, and focusing on three technologies currently being debated: forensic DNA phenotyping (FDP), forensic genetic genealogy (FGG), and forensic epigenetics (FEpi). In this endeavour, the concept of the “social life of DNA,” proposed by Corinna Kruse [18], which captures the different social relationships observed in the processing of DNA as it moves from crime scene to courtroom, resonates. As such, we analyse the different life stages of advanced forensic DNA technologies. While DNA profiling is a relatively widely accepted technology, having stabilized and been implemented across different contexts, the three technologies we focus on are yet to reach such a point of stability. In reviewing these examples, we highlight key ethical concerns to assist forensic practitioners in expanding their anticipatory capacity and thus work towards ethically sound decision making. By anticipatory capacity, we refer to antecedent considerations in science, and in real-world application of science, set within a wider context of liberal-democratic social and cultural norms and values. These three technologies were selected, firstly, due to controversy concerning their utility and reliability/validity specifically, and their ethics more generally [19,20]. Secondly, these technologies present opportunities to engage with ethics as lived practice, as they pose ethical dilemmas. Thirdly, debates are ongoing over their use, and measures in place to regulate these technologies differ across countries. Finally, each of these technologies is currently at different life stages. Forensic DNA phenotyping, although still the subject of complex politics of legitimation and contestation [20,21,22,23], is currently regulated and applied in some countries such as Slovakia and Germany. Forensic genetic genealogy is under intense scrutiny in many jurisdictions and upcoming regulation can be expected (see for example recent initiatives in the USA and in Australia). Forensic epigenetics is being developed and has not (yet) been considered widely with respect to regulation and/or application.

## 3. The Limitations of “Thin” Ethics in Forensic Genetics

Controversies surrounding advances in forensic genetics could (at least in part) be a consequence of a structural deficiency in comprehensive, systematic understandings of what ethics is—or should be—in forensic genetics and related technological innovations. Let us explain what we mean by a “thin” understanding of ethics.

Firstly, forensic geneticists will encounter ethics as part of an administrative process to be followed to enable research, innovation, and implementation: following codes of practice, completing forms, attending training, and submitting authorizations to some form of the institutional oversight body. Such administrative tasks and tools are clearly essential in ensuring forensic scientists navigate the most basic rules of research and technology use. However, the main thrust of this ‘procedural ethics’ [24] is to comply with administrative and legal procedural requirements, with ethics conceived of as ‘passive’, lending itself to the interpretation of ethics as a hurdle to research, innovation, and (positive) societal impact that forensic genetics could otherwise make. Consequently, exclusive adherence to this “thin” procedural ethics can extinguish the light that a (pro-)active approach can shine on the complex environment within which forensic work takes place. It also ‘disengages’ forensic geneticists (both in research and casework) from ethics to the detriment of forensic genetics, and more widely to science in the criminal justice system.

Secondly, within this “thin” understanding of ethics, ‘privacy’ tends to be perceived as an interpretable legal concept, addressed by compliance with the law. Moreover, ‘data protection’ is often used as a proxy for ‘privacy’, whereby observance with principles of data protection will suffice, yet ‘privacy’ is far more expansive and manifold than mere data protection (pp. 33–50, [25,26]). A reductionist understanding of ethics as privacy obscures other vital aspects of ethical practice, including attending to justice and fairness, human dignity (and its cultural values, see, e.g., recommendations by the Law Commission of New Zealand vis-a-vis forensic genetics and the Māori), and the physical and informational integrity of a person and/or a group of persons (community). A focus on privacy also misdirects debate into a cul-de-sac of arguments over balancing ‘personal privacy’ with ‘public security’ since this dichotomy is erroneous; privacy and security are symbiotic [27,28].

Thirdly, technology may be perceived as a mere tool, process, or method. However, as several years of consolidated scientific research have shown (particularly in social studies of science and technology in health research, and more recently in the social studies of forensic science), technology is a product of social, cultural, and moral facets of society. Technology is always technology-in-practice [29] resulting from, and shaping social interactions and the organisation of social life. As Greenhalgh and Swinglehurst (p. 3, [30]) suggest, considering technology as social practices “moves us on from studying either people or technologies” towards a rich, inclusive analysis of technology as an aspect of social life, considering technology-in-use as well as the discourses and practices of technology [31].

Reducing technology to its instrumental aspects conceals many ethical decisions that shape technology, from its conception to application and maintenance as well as to its oversight. Thus, we consider “thin” or ‘procedural ethics’ insufficient in forensic research and application, with privacy as just one ethical concern. Nor is technology a mere tool or method but should be viewed as social practice. Thus, we propose an approach to ethics as intrinsic to the lived practice of forensic geneticists, making deliberate decisions about daily practices and procedures, acting in light of these decisions, while continually reflecting upon their broader impacts now and in the future.

## 4. Ethics as Lived Practice

Ethics as lived practice extends throughout the life cycle of science: from development and implementation to regulation and oversight. Regardless, then, of the role(s) that professionals fulfil—whether as research scientist engaged with the laboratory-based development of forensic applications; as a forensic practitioner working on criminal cases; as an expert advising on casework or giving evidence in court—ethical engagement is inalienably part of professional practice. Such ethical engagement is inescapable because (i) the development of forensic technologies will affect human beings; (ii) those conducting forensic casework, and working with forensic data and information, are, as all other human beings, subjective, open to influence, and fallible; and (iii) forensic casework builds upon and contributes to selecting, categorising, and curating data and data repositories as reference points, thus making choices that will affect future forensic research and casework, and as such, many more humans.

Our understanding of ethics as lived practice stands in contrast to the notion of ethics as a necessary evil and to the “thin” ethics discussed above—where ethics only arise at bounded points in time when the ethics ‘hurdle’ is jumped. Ethics as lived practice, instead, recognises that decision-making is ongoing, and an intrinsic part of the practice. Continual reflection, then, becomes central: taking time, sometimes briefly, sometimes for a longer period; alone or as a team; discipline; or community, to (re)consider the basis and motivations on which decisions are being taken, as well as their consequences. This also calls for serious engagement with others, including those from outside the laboratory (such as criminal investigators, social scientists, policymakers forging legislation and regulation, as well as the civil society), to discuss research, analysis, and future application, within their wider social and cultural contexts. We call upon the forensic genetics community to build on emerging strengths here by making ethical reflection—as a way of building anticipatory capacity—an integral part of forensic genetics work.

The remarkable potential of forensic science, to not only contribute to the pursuit of justice, but to shape and direct society, is highly valued, but comes with enormous responsibility. Acknowledgment of the wider impact of forensic work means that ethics ought to be reflected in scientists’ and practitioners’ training, in their daily decision-making, and in contributions to scientific and public debates about forensics. Guillemin and Guillam [24] explain that ethics work occurs in researchers’ daily practice beyond procedural ethics. Such a practice-related understanding of ethics has been suggested as a way of incorporating principle-based reflections about the impact and implications of scientific design and work into everyday decisions [32]. Furthermore, Guillemin and Guillam suggest that researchers may encounter ethically relevant moments, where the approach taken or decision made has important ethical ramifications, but where researchers do not necessarily feel themselves to be on the horns of a dilemma. In fact, in some cases, it might be clear how the researcher should respond or proceed, and yet there is still something ethically important at stake (p. 265, [24]). While the scientist or practitioner might not perceive this moment as particularly vital or difficult, the decisions made at this moment can have significant long-term implications and impacts: on the person making the decision and their forensic specialism; on forensic science more widely and its perception among users, policy and publics; on the domains using forensic science services; and, via these domains, on ethical issues such as: social justice; equality; equity; dignity; privacy; integrity, and so forth. In these ‘ethical moments’ (for a look at ethical moments in forensic genetics, see e.g., [33]) different perspectives are brought together, to identify and resolve disagreements, making visible ethical reasoning about scientific practice and the use of scientific knowledge [20].

Consider, for example, the forensic scientist tracing a partial DNA match indicative of a familial relationship between a suspect DNA profile and that of a dragnet/mass test volunteer. If permitted by law, the modus operandi is clear, and the practitioner should always follow what the law prescribes. However, the law does not, and cannot, provide for every situation practitioners will come across in their daily practice. If the law has not explicitly provided for a situation, there is a significant decision to be made: should the expert report a partial match and thus a potential familial relationship, even though familial searching is not explicitly regulated in their jurisdiction? In pursuit of a perpetrator, it may be legitimate to report, even if such analyses are not formally accepted practice (e.g., not regulated by law, nor a laboratory accredited for), thus shifting the ultimate ethical decision to others, such as the investigating judge (in predominantly inquisitorial systems) or the police and prosecution (in more adversarial systems). Nonetheless, the forensic geneticist is required to make an ethical decision about their practice at this point, and procedures to support them in doing so—perhaps by calling on them to reflect upon their decision—may often be non-existent or very limited. This example, like many others occurring daily in forensic genetics, clearly demonstrates that ethics cannot be limited to an initial passing of a procedural ethical hurdle, but relies on many decisions around reliability, utility, and legitimacy [20] that an individual or their team take in forensic research and casework.

Ethics as lived practice demands more attention, while a “thin” understanding of ethics, anchored, for example, in ‘procedural ethics’, is insufficient for responsible (and as such also accountable) forensic genetics practice. We argue that ethics as lived practice, therefore, means that forensic researchers and practitioners identify and understand the need to address opportunities as well as limitations of their work set within an institutional and societal context—e.g., in criminal justice, migration management, or disaster victim identification. In the next section, we discuss ethical decision-making in forensic genetics practice using the example of data repositories because their creation, curation, and use are vital to forensic genetics research and can play a significant role in investigative work and law enforcement.

## 5. Building upon Strong Foundations: Ethical Decision-Making in Genetic Databasing

Central to forensic genetics is the data collected from individuals from diverse groups and curated for use by the scientific community. However, as we show in the example of forensic genetic genealogy below, data collected and curated for non-forensic purposes are also increasingly of interest to forensic geneticists and law enforcement agencies. Data collection and curation in repositories form a key life stage of genetic data and ethical considerations in forensic genetics, as with any human genetics field, therefore begin with the sampling design for DNA data for research and for databases such as EMPOP or YHRD. Since some forensic genetics services depend on population data, genetic data from diverse populations have been widely collected from individuals across the globe for decades. Since the 1970s, the fields of human and medical genetics have developed and adopted a number of ethical standards and procedures, including, most notably, requirements for informed consent and approval by ethics committees [34]. Such processes have been continuously refined and upgraded, and yet ethical debates around genetic data continue [35,36,37,38,39].

In contrast, the forensic genetics community’s leading journal, Forensic Science International: Genetics, introduced statements about these basic ethical standards (informed consent and ethics approval) as a publication requirement as late as 2010 [40]. In 2020, guidelines were updated and extended, with a specific focus on population data from minorities or indigenous communities [38]. Until 2010, there were no ethical standards for authors to meet when publishing DNA data; and data collected before 2010 without informed consent and ethical approval is still used, shared, and uploaded. Furthermore, forensic genetic databases usually require the publication of data in peer-reviewed journal articles before such data can be uploaded, along with ethical approval and/or evidence of informed consent. However, in some databases, data can be uploaded without meeting these criteria [41]. Notably, such databases are not only used for research, but also for investigation by law enforcement authorities. Ethical debates around these databases have only now begun.

In human genetics, the demands of obtaining consent for samples from sufficiently large numbers of individuals requires the building of an ethical regime of institutional and personal infrastructures in order to enable the legal and ethical use of samples and profiles (e.g., [42]). Numerous ethical decisions, consciously or not, are inevitably made when recruiting donors: whom to include or exclude, how to frame the research to donors about why samples and data are requested, what to give back to donors and their community, and so forth. In forensic genetics the stakes are higher, as persuading individuals to donate their DNA is challenging. Willingness to donate DNA for forensic purposes used to be considerably lower than for health purposes [43]; albeit more recently there has been an increase in voluntary offers of existing genetic profiles or donation of new genetic data to enable the forensic use of ancestry, health, and research databases [8]. Such donations place a particular responsibility upon researchers and analysts to be very clear about the potential uses of such data, not only at the time of sample collection but also afterwards, as data moves along different life stages.

For such data repositories to serve law enforcement, data acquisition needs to be as extensive as possible to represent diverse populations. Previous resolutions to this problem have attracted criticism: E.g. some forensic practitioners have argued that overrepresentation of marginalised minorities on databases means they are more likely to be exonerated, but this theory has yet to be supported with evidence while emerging database uses for biogeographic ancestry testing and capacities to de-anonymize profiles increase the riskiness of overrepresentation [12]. Another example is that in China, but also in numerous other countries, police and security forces have been involved in collection efforts, nullifying any supposed ‘consent’ [1,41,44,45]. Recent discussions focus upon the reuse of data and samples previously collected under unclear ethical conditions, or without any demonstrable adherence to ethical principles [1,12]. Some of these datasets have been obtained from vulnerable populations and shared by, and between researchers from forensic, medical, and population genetics, raising serious questions about whether such data transfers violate general ethical standards.

Even if data have been sampled ethically and safely stored in databases, to be used as training and test data in calibration exercises, or as reference data for criminal investigative application, social and ethical issues may still arise. A key issue is whether the data is representative of the ‘target’ population [1]. Yet rarely discussed are potential problems arising from non-representative data collection. For example, if the training data for an FDP application has been sampled too selectively (e.g., [46]) and hence is very homogeneous, probability calculations might be overly optimistic. Then, in a criminal investigation, the technology will overestimate its own reliability against the backdrop of a local population that is not as homogeneous as the training data. In biogeographic ancestry testing (BGA), so-called “mixed populations” in training and reference data lead to erroneous test results and overly optimistic reliability estimates [47]. However, “mixed ancestries” will most likely be more commonplace in real life but, in a worst-case scenario, the technology will produce erroneously high probability estimates, and yet these are more likely to be trusted than other evidence-based on perceptions of scientific objectivity and DNA’s standing as “biological witness”—and thus risk leading investigations astray [45].

It is critical, therefore, that ethics as lived practice commences at the very first life stages of DNA data: when collecting and curating data and establishing repositories. As this DNA data is then utilised by forensic practitioners within the context of the criminal justice system, the next vital life stage of forensic genetic data, further ethical decision-making is required.

## 6. Challenges of Context and Communication: Ethical Decision-Making as Forensic Expert

Forensic DNA technologies have been framed by metaphorical affirmations of DNA as representing a “truth machine” [48] or the “gold standard” of evidence [49], reflecting high expectations based upon unrivalled (alleged) levels of validity, certainty, reliability, and objectivity. Besides disregarding issues arising during the “social life of DNA” [18], as it moves from crime scene (or database) to court, such metaphors eclipse the challenges of communicating probabilistic DNA evidence to non-experts [50,51]. These challenges occur at many different contact points: in the regulatory process leading up to new or amended laws or regulations; in the training of geneticists and investigators; in public communications during police investigations; during and after a trial; and in the aftermath of a solved case when the efficiency of different investigative tools is compared and highlighted. In addition, as DNA analysis becomes more sensitive (entailing the analysis of mixed traces, DNA transfer, etc.), and more open to interpretation, it becomes yet more challenging to report [52], with scientists deciding whether to apply a probative value to a result, or provide an inconclusive result that neither includes or excludes the suspect [53].

A rational assessment of what advanced DNA profiling techniques can bring to a police investigation is essential but rare. For example, in the UK, ‘familial searching’ has been available since 2012 and utilised in 120 cases, with just nine reported ‘successes’ [54], perhaps challenging proportionality equations over its use. A realistic appraisal must particularly consider how ‘results’, once outside of a laboratory or research setting such as in forensic validation, are applied during real-life criminal investigations. A host of confounding variables and diverse considerations come into play during the criminal process, which may distort or nullify laboratory results: “while a forensic technique may have a valid scientific basis and prove reliable in a laboratory setting, how can it be ascertained that it remains reliable when applied in the present case? What are the error rates associated with the technique, and are there any pertinent factors that could jeopardise the reliability of the testing in this particular instance?” (p. 240, [13]). A forensic scientist engaged in ethics as the lived practice may be required to inform these more ‘realistic’ appreciations of what forensic genetics can and cannot achieve within the context of police investigations, particularly when asked to deploy advanced techniques. Both law enforcement and the public should not be ‘oversold’ benefits that may never be realised outside of a research laboratory [33,55]. Our three case studies below address intelligence-related uses of forensic genetics in investigations, when arguably, the dangers of over-selling capacities by forensic scientists and service providers as well as the ready acceptance of such leads by investigators in order to progress a complex case, are high.

Assuming that DNA science moves from investigation onto a later life stage: forensic intelligence from genetic genealogy, but perhaps also from other techniques, may lead to requests by police for arrest warrants. This potential outcome would require such forensic intelligence to be carefully articulated and to take into account the context in which it may be used, including institutional policing cultures and wider political debates, e.g., about migration and criminality. In inquisitorial systems (but highly unlikely ever in adversarial systems), forensic intelligence may even make it into the report submitted to the investigating judge. At some point, if rarely, forensic experts may be called as witnesses to explain the intelligence.

We know for forensic evidence that the judge may be required to take on a “gatekeeper” role to ensure that “junk science” remains inadmissible at trial. This role and its fulfilment has been highly criticized already for adversarial systems where mechanisms are in place for contestation [56,57], with conclusions drawn that “scientific illiteracy on the part of the legal profession, when coupled with the flaws in forensic science, forms a ‘toxic combination’” (p. 82, [56]). The trial process rarely affords even the scientifically literate much opportunity to interpret and apply ‘correctly’ complex tests (assuming there is a ‘correct’ application) for whether scientific evidence is ‘sufficiently reliable’ [58]. In addition, such admissibility testing may come too late in the criminal process. Consider, too, that in the inquisitorial system forensic expertise is very rarely challenged. Even where challenges to scientific information exist, decisions to admit such information at trial can appear arbitrary, or inconsistent, with decisions based upon non-scientific criteria [59].

Scientists, in providing intelligence as well as in giving testimony, are called upon to make ethical decisions about which information to present and how it should be best communicated. Much work has gone into, e.g., developing verbal scales as equivalents of reporting numerical values and effectively interpreting probabilities [60,61,62]. Forensic geneticists’ adherence to ethical principles and practices in giving testimony, is a key aspect of their responsibility as practitioners (see, e.g., [63]), yet can be complicated by different scientific, investigative, and juridical pressures and demands on the scientific expert. A key responsibility is avoiding ‘over-egging the pudding’. As such, scientific experts have personal and disciplinary responsibility for their communication, and thus at least in part, a wider responsibility for the role of intelligence in investigations and for scientific evidence in court.

In response to the exposure of mistakes, misrepresentations, and misunderstandings of DNA evidence that have led to failed investigations or miscarriages of justice, there have been efforts to regulate forensic science [64]. Political crises induced by a wrongful conviction or scandal have most often proven necessary to provoke regulatory intervention, but this may come too late for an individual who cannot be adequately compensated for their loss of liberty or diverse publics whose trust in the administration of justice is irreparably damaged. In addition, the implementation of such regulation, including the writing of standards and operating procedures not underpinned by rigorous research, has been criticised [65]. It also remains the case that error rates and limitations of techniques often remain sufficiently unarticulated, or even undetermined. Yet even assuming extra resources are provided to produce robust error rates, it is questionable whether risks can be meaningfully quantified, given the inherently contextual nature of forensic DNA evidence. As we argue, many questions are not purely scientific but include ethical considerations from the very outset of the social life of DNA. Acceptability of DNA evidence thus turns, in part, upon the criminal justice system’s values and public tolerance of error, but classic regulatory risk analysis typically takes little (if any) account of sociological, economic, ethical, or even legal considerations. The countervailing difficulty of encouraging innovation and development within regulatory parameters, demands innovative solutions operating within an ‘ethics as lived practice’ framework. This will require embedded ethical practice, as well as regulatory flexibility and pluralism, to avoid stagnation, and setting forensic science ‘in aspic’ [66].

Such flexibility to innovate is especially critical when also facing the challenges of commercialization, leading to the confluence of competition and profit-oriented practices with those of scientific endeavour and of the criminal justice system. This confluence can lead most obviously to conflicts of interest, which can affect the portrayal of validity/reliability and utility of forensic technologies, but also directly impact the quality of forensic analyses undertaken; the reporting of these forensic analyses, and also the scientifically robust development of existing and emerging forensic technologies [67,68,69]. Forensic scientists constantly make ethical decisions in this context, as discussed above for scientific and evidentiary reasons. In the commercial context, additionally, forensic practitioners make ethical decisions around accreditation for services; negotiating access to contracts; customer satisfaction; access to material and financial means in order to conduct research; contestations around other service providers, etc. [33]. Many scientists work together with companies in developing forensic technologies, equipment, and also at times, services. They may be delivering services for commercial providers, or they may be in direct competition. Due to the potential conflicts of interest, the commercial context makes it even more important for forensic scientists and practitioners to reflect on their decision-making in order to ensure responsible and accountable delivery of forensic genetics analyses and interpretations.

## 7. Case Studies

Our call to adopt ethics as lived practice in decision-making processes throughout key stages in the life of forensic DNA is underscored by three case studies. These vignettes explore ethical decision-making in forensic DNA phenotyping, forensic (or: investigative) genetic genealogy, and forensic epigenetics. The three fields share a common basis: they are presented by forensic scientists and users as providing intelligence for investigations without apparent leads by generating data on a group of suspects, and they are portrayed to facilitate even cold case investigations, based on probability calculations and human (epi)genetics.

### 7.1. Forensic DNA Phenotyping (FDP) and Biogeographic Ancestry Testing (BGA)

FDP and BGA have provided intelligence to criminal investigations that, at times, have resulted in the detection of suspects. To date, in the USA, hundreds of investigations have drawn on intelligence from FDP and BGA analyses since 2015. In Europe, these techniques (especially BGA), have found application in cases since at least 1999, including in the Marianne Vaatstra case [70]; the investigation of the Madrid train bombings of 2004 [71]; the Phantom of Heilbronn investigation in 2008 [72]; and the detection of the murder of Eva Blanco Puig in Spain in 2015. In the Vaatstra case in 1999, an early form of BGA was used to the effect of taking public pressure off asylum seekers who had been considered prime suspects. In the Madrid investigation, the BGA analysis contributed some intelligence to the investigation. In the case of the Phantom of Heilbronn, the results did not contribute to the detection of perpetrators; rather, they seemed to fuel the bias of investigators. While the case of the Phantom was widely debated, unsuccessful applications of FDP or BGA tend not to be discussed publicly. Yet insight into limitations can assist the wider forensic genetics community of scientists and users, but also the public, in understanding the effective parameters of these techniques.

A more recent case is the rape and murder of Maria Ladenburger, on 16 October 2016, in the German city of Freiburg im Breisgau, Germany. At the start of the investigation, suspicion quickly fell on the large migrant community of approximately one million people who had come to Germany in 2015/16. Representatives from politics, policing and forensic genetics suggested that expanded DNA technologies would contribute to faster detection of the perpetrator if only analyses of the perpetrator’s (1) appearance, using FDP, and (2) continental origin, via BGA, would have been permitted. While many German experts, including the German Spurenkommission (trace commission), initially argued that the Ladenburger murder would have been an ideal test case for application of FDP in a criminal investigation, senior German forensic geneticists working on FDP later commented that this specific case would not have benefitted from its use [73,74,75]. Yet, the raised expectations of audiences and investigators remained unrealistically high, leading to the legalization of FDP in Germany in 2019. Yet the Maria Ladenburger case was solved through traditional police investigative techniques, resulting in the arrest of Hussein Khavari, an Afghan migrant, previously convicted in Greece.

The discussion of FDP and BGA, in this case, has raised several questions, specifically around the communication of the utility and reliability of these emerging technologies, but also about the implications of using technology that aims to identify categories of people rather than individuals [20,76]. Overall, FDP and BGA raise a number of ethical questions for forensic scientists, questions that need to be reflected upon time and again in order to deliver responsible and accountable science. While we focus on the estimation of externally visible characteristics in this case study, we do acknowledge that the testing of biogeographic ancestry (BGA) has its own technical and ethical pitfalls, some of which it shares with FDP and some of which are specific to BGA. For example, there are significant incongruities between ‘population’ labels used by the forensic laboratory and those used by police forces which risk miscommunications, while opening up casework to being highly susceptible to bias (e.g., [72]).

First, FDP works on the basis of attribution rather than identification. However, defining visible/external attributes, e.g., hair/eye/skin colour, must be couched in ambiguity because they are subject to individual evaluation, e.g., colour perception, but also culturally informed perception of such traits (see further [15]). These individual attributions are then compared to non-uniform standards. Here also, as mentioned earlier, the choice of reference data has a fundamental influence, requiring forensic scientists to reflect on the biases that go into defining reference data. In many cases, DNA analysis results might be meaningless or even misleading. Articulating such crucial limitations and distinctions to users is central to ensuring the reliability and utility of FDP in an investigation, even if reporting forms seem to leave scant space for such important information. Space and time must be made for communicating these aspects since they can significantly impact the understanding investigators develop of intelligence generated via FDP analyses.

Second, the investigative value of forensic genetics arises from the capacity of technology to either deliver evidence or intelligence to detect crimes and perpetrators. In the case of FDP, intelligence is produced only in as far as the genetic analysis can either point towards (or away from) a group of shared attributes/characteristics a perpetrator may have. FDP is about shared characteristics and as such applies suspicion to groups, collectivising suspicion [77]. However, since FDP deals in probabilities, the extent to which a group can be excluded can never be 100% certain. A key issue here is that the technology works best on minority groups (of an appearance distinctly different to that of a majority population) to deliver on its aim of directing the next steps in an investigation (e.g., [78]). It is thus vital to reflect when using technologies such as FDP, on their investigative value, which is reduced when the analysis points towards a majority group as then the expected ‘reduction’ in suspect populations is not achieved. Effectively, this may lead to minority groups being implicated in active investigations more often because the investigative tools work less well in the case of majority populations [79]. At the same time, forensic technologies such as FDP do not work in isolation from policing practices and institutional cultures [21]. Therefore, forensic scientists need to reflect on the discriminatory power of the technology that lies at its heart-both at policing and at public levels. This was the case in the Ladenburger investigation where non-European migrants were associated with criminal behaviour because of the way that FDP was, for a long period of time, communicated as the key technological solution to violent crime investigations.

Third, and this may be the most apparent element to forensic scientists, FDP remains an emerging technology. While pigmentation analyses may be well advanced for darker and lighter aspects of the continuum, medium levels of pigmentation remain more difficult to test for reliably (e.g., [80]). Furthermore, public debates between politicians, police, and scientists around the Ladenburger case confused the technology readiness level of FDP: while scientists were talking about lab-based levels of testing and aspirations for future analytical capacities, perhaps even up to “facial composites”, publics, politicians and police representatives perceived of these capacities as already realised and ready for use. Key messages around the reliability and utility of FDP were mixed up. This may yet result in a significant loss of trust in the technology and forensic science in policing should its legalisation in Germany in 2019 not lead to any significant positive results in investigations in the near future.

### 7.2. Forensic Genetic Genealogy (FGG)

From 1974 to 1986 in California, USA, crimes including murder and rape, were committed by the same unknown individual. In 2018, investigators uploaded genetic information derived from crime scenes onto the public-access genetic ancestry database, GEDmatch. This database aggregates genetic information from citizens who upload the results of their direct-to-consumer genetic tests from genealogy companies, such as AncestryDNA, 23andMe, MyHeritage, and FamilyTreeDNA. Investigators found partial DNA matches assumed to belong to distant relatives, which enabled the creation of a ‘family tree’, primarily based on online genealogy database records, and including information from sources such as social media and other records. The data of these records sit outside of the forensic domain, and their use in policing raises a number of legal and ethical questions, including on the level of appropriate consent; dual/unintended use of data; and whether the capacity exists or can be introduced for blocking or deleting data from such sources if data subjects disagree with law enforcement uses of their data. Consequently, Joseph James DeAngelo, 72 years old, was identified as a suspect, and his “abandoned” DNA collected to facilitate confirmatory DNA analysis. Although the Golden State Killer was not the first case in which this technique has been used, it has become what Prainsack and Toom [81] term a “founding myth” and is being enthusiastically explored by police globally, and several cases have now been solved through FGG internationally. According to Kling et al. [82], FGG has been used to generate investigative leads in nearly 200 cold cases and some active investigations (see also [10,83,84]). In Sweden, prosecutors have allowed investigators to use consumer genealogy databases to solve cold cases: missing persons and a criminal case [85]. In Canada, two cold cases have been solved through forensic genealogy [8]. In the UK, a study has assessed the likely effectiveness of long-range familial searches through GEDmatch [86]. The Golden State Killer also sparked interest in this procedure among citizens and the companies offering these services, with existing databases now allowing for and, in some cases, encouraging law enforcement searches, as in the case with GEDmatch (recently acquired by commercial forensic service provider Verogen which raises further ethical issues which we do not have the space to explore here) and FamilyTreeDNA. In addition, services exclusively dedicated to law enforcement searches, e.g., DNASolves, DNA.Land, Geneanet, and Geni, are emerging [82] and some law enforcement agencies are setting up their own FGG services.

The ethical implications of FGG are wide-ranging. First, this technique increases acutely the amount of information that can be garnered from DNA data, as it makes use of a significant number of markers and uses SNPs, instead of a relatively reduced number of markers and STRs, as used in ‘traditional’ forensic DNA testing and databases. Genetic genealogy also allows users to identify distant relatives [34,35] and disease predispositions.

Second, FGG challenges the notion of what constitutes a DNA database used for law enforcement. Until recently, forensic DNA databases were solely constituted of DNA profiles of individuals with some degree of involvement with the criminal justice system [36]. Yet suspects are now being searched for in databases consisting of voluntarily uploaded genetic data from citizens who wish to know more about their health, ancestry, and/or search for relatives: a clear example of ‘function creep’. Such repurposing of DNA data requires a discussion of (informed) consent [33], data ownership, and the accountability of genealogy companies and law enforcement in relation to human rights.

Third, the genetic databases being used/created for FGG usually have a significantly different composition than “traditional” forensic databases, the latter over-representing groups that are most likely to interact with the criminal justice system, such as racial and ethnic minorities [87] while genetic genealogy databases are mainly composed of an economically privileged and a European-descent population [51]. This leads to the expansion of affected populations as involvement with the criminal justice system is no longer a prerequisite to inclusion in law enforcement searches and follows broader patterns of surveillance expansion that encompasses ever more citizens. FGG, therefore, brings individuals whose profiles are in a recreational DNA database, under the ‘genetic surveillance’ gaze, making them suspects by association [88]. This ‘function creep’ illustrates the potential for wider ramifications of decisions that are particularly risky when impacts cannot be known and their diffusion across society remains uncharted. However, it should be noted that different databases have varying policies in terms of making genetic data available for law enforcement searches [89] GEDmatch and FamilyTreeDNA encourage such searches. GEDmatch offers users to opt-in while FamilyTreeDNA, in 2019, instituted an opt-out policy that automatically opens up users’ data to law enforcement searches. However, under the jurisdiction of the European Union’s General Data Protection Regulation (GDPR), users from current and recent European Union member states who have profiles on FamilyTreeDNA have been automatically opted out. This means that European Union citizens must opt-in if they wish to have their DNA profile included for criminal investigation purposes [8,54]. Adopting a different stance, companies such as AncestryDNA and 23andMe have categorically denied access for law enforcement agents to such data, unless subpoenaed by a court order to do so [89].

### 7.3. Forensic Epigenetics/-Genomics (FEpi)

Imagine: A woman is found dead, and DNA samples collected from crime scenes do not provide any investigative leads. After making a genetic estimation of appearance traits and biogeographic ancestry (see case 1), it is decided to conduct epigenetic prediction of chronological age [90], as well as epigenomic prediction of lifestyle and environmental factors through the DNA samples. Results indicate that the presumed suspect is approximately twenty years old, smokes tobacco, has a high alcohol intake, and probably belongs to a specific “socioeconomic status”(p. 7, [91]). Based on these results, police investigative work will be prioritised towards a more specific suspect pool. While hypothetical, this is the aspiration of forensic epigenetics.

Until recently, despite the enormous interest in post-genomics research, forensic applications of epigenetics were mainly restricted to body fluid identification, differentiating between monozygotic twins, and age prediction [90,92,93]. The potential uses of epigenetics in forensic research have only recently expanded, with expectations that the understanding of how environmental factors impact our DNA might be useful ‘intelligence’, by identifying suspects’ lifestyle, such as smoking habits, alcohol intake, other drug use, diet, body shape and size, physical exercise, zone of residence, and socioeconomic status [91,94]. As with FGG, the potential ramifications of FEpi are extensive and impossible to fully predict, when technology is still in an embryonic life stage. It is thus critical that potential consequences are contemplated to develop anticipatory capacity.

First, revealing lifestyle and environmental characteristics directly interferes with the right to privacy of individuals and populations. Despite arguments—already expressed and disputed [15]—that “the genetic prediction of obvious appearance traits [such as smoking] because their external visibility cannot be considered private” (p. 8, [91]), it is clear that inferring smoking, drinking and drug and food consumption habits indirectly correlates with health information and medical conditions, further blurring the distinction between forensic and medical uses of genetic data.

Second, we might question how useful information about smoking and drinking habits might be for investigators since the indication that a suspect smokes and/or consumes alcohol does not significantly advance a criminal investigation without any suspects. Similar to FDP (case 1) and FGG (case 2), FEpi casts suspicion across entire groups by clustering ‘suspect’ populations that share certain lifestyles.

Third, epigenetic markers are, to a considerable extent, established in early development, potentially transmissible to subsequent generations, and sensitive to environmental factors and lifestyles [95,96,97]. If we look back to the hypothetical criminal case and focus upon the information that the presumed suspect is approximately twenty years old, smokes, has a significant alcohol intake, we must take into consideration (i) that maternal smoking and alcohol consumption during pregnancy could cause epigenetic changes in the offspring; (ii) passive smoking also might impact the epigenome; (iii) alcohol-dependent epigenetic signatures are partly reversible upon abstinence [91,94]. This suggests that the variability of epigenetic markers does not constitute reliable or consistent predictors for identifying suspects. Although FEpi is still being developed in terms of scientific validation, an approach that is anchored in ethics as lived practice implies that issues around reliability, utility, and legitimacy [20] must be considered focal points to be addressed and discussed.

## 8. Concluding Remarks

As science is afforded a particularly powerful role in knowledge production and meaning-making in the criminal justice system, could this lead to a ‘technological tyranny’ [98] in which individuals and communities have to prove their innocence against probabilities? Might forensic scientists inadvertently contribute to creating new kinds of inequalities and insecurity when it comes to policing and criminal justice? (p. 365, [99]). In response to such questions, particularly facing forensic geneticists, in this paper we propose a move from the dominant “thin” ethics approach, to one of ethics as lived practice. We argue that forensic scientists are ethical agents, that is, they make ethical decisions daily and not just at procedural moments such as requesting ethics approval or reporting to an institutional review board. Rather, they have the agency and power to make decisions with far-reaching impacts. Such decision-making can be more apparent at some points than others and must go some way to building ‘anticipatory capacity’ for the governance of reliable and responsible forensic genetics. In this paper, we have articulated some of the key stages in the life of forensically relevant (epi)genetic data at which ethical decision-making takes place, and where it is vital to take the time to reflect on ethical principles and practices. Our aim has been to raise awareness of the need for continuous reflection upon ethical decisions and to articulate some of those key instances in which such reflection is clearly vital.

Key lessons:Anticipatory capacity and ethics as lived practice are vital components of the work of forensic scientists. This implies the need to be able to anticipate how technology use will impact individuals and groups of people potentially drawn into investigations, as well as wider community relationships, and criminal justice more generally. Our first key point is that forensic geneticists need to engage with ethics in all stages of the sample and data life cycle.There is an urgent need to address the power of forensic (epi)genetics to cast suspicion over a wide range of individuals and communities who may share visual characteristics, common family ancestry, and/or lifestyle habits. This poses several issues that forensic genetics should be aware of when developing and/or applying such technologies, as we move towards a society where individuals and even communities are increasingly called upon to prove their innocence. Our second key point, therefore, is that forensic geneticists need to be aware of the active role that their research and technologies can play in either strengthening or threatening such fundamental legal principles as the presumption of innocence, equality of arms, and the legal burden of proof.Our third key point is that clarity and open communication about real-world capacities of technologies such as FDP, FGG, and FEpi (as well as BGA which is not discussed in depth in this paper) must be at the core of discussions around their use. Their reliability and effectiveness outside the laboratory, and their utility for specific cases, must be reflected upon, and openly and clearly articulated to users and commissioners of such technologies in order to build legitimacy, including public trust, responsible innovation and practice, and accountability.

## Data Availability

Not applicable.

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
