# Peer review of "Ethics as Lived Practice. Anticipatory Capacity and Ethical Decision-Making in Forensic Genetics"

_genes, 2021, doi:10.3390/genes12121868_

Round 1
Reviewer 1 Report
The paper is very interesting.
The authors make a critical analysis of the ethical issues on the application of DNA analysis in the forensic field. In particular, the paper describes three examples (DNA phenotyping (FDP), forensic genetic genealogy (FGG), and forensic epigenetics) that provide the reader with an important point of reflection.
However, there seem to be several concerns in this manuscript. The paper will be improved when the authors revise them according to the following comment:
- No valid solutions or alternatives to the use of these methods are proposed. The authors explain why it can be ethically "harmful" to use some technologies, but do not report any alternative solutions and the advantages that the use of these technologies. Indeed, it would be interesting to compare the ethical risk of the application of these technologies and the actual benefit that these technologies bring to the resolution of criminal cases.
- Authors should insert in the manuscript the advantages that forensic genetics provides to society.
- Several concepts in the paper are redundant.
- The references in the paper should be reviewed.
Author Response
Dear Professor Morling, dear Ms. Shen, dear reviewer,
Many thanks to you for your time reading and commenting upon our manuscript. We appreciate the opportunity to revise our paper. We have made revisions in line with comments, and present our responses here in the format <reviewer’s comment> / our response. Here, we include new references in Harvard style for ease of identification, but in the manuscript we apply the journal’s endnote approach. We include all key responses as tracked changes in the manuscript upload. We look forward to hearing from you.
<The paper is very interesting.>
Many thanks for this supportive assessment.
<No valid solutions or alternatives to the use of these methods are proposed. The authors explain why it can be ethically "harmful" to use some technologies, but do not report any alternative solutions and the advantages that the use of these technologies.>
Thank you for this comment. We take seriously the interest of the forensic genetics community in learning about how to build upon social scientific insight for ethical deliberation in forensic research and practice. The ethical approach we propose in this paper (“ethics as lived practice”, as we call it) cannot and does not aim to propose or dictate technical or legal solutions. We believe that kind of approach needs to be a result of a joint effort between a plurality of perspectives engaged with forensic genetics, including, but definitely not restricted to, social scientists, ethicists and legal experts as ourselves.
Instead, the aim of our paper is to offer an approach to how the community must reflect on their daily decisions in both scientific research and case work as part of their professionalism and ethical practice. This is a vital resolution for scientists and case workers, and calls for/requires a reacquainting with ethics by these groups away from a “thin” understanding of ethics as discussed in more detail in our paper. We do so in part by discussing what ethics is about, and in part by providing illustrative examples for current and emerging technologies.
Finally, we would like to point out that the paper does not include a comprehensive evaluation of the ‘harms’ or ‘advantages’ or otherwise of these technologies, and does not attempt to engage in any ‘balancing’ or weighing up of different techniques/technologies versus others, so it would be outside of the purpose of our paper to report upon any ‘alternative solutions’ to forensic genetic technologies. We have however, in response to this comment, added a short section by way of some clarification.
In the Introduction (page 2), we have added the following:
“In some parts of the world, forensic genetics research and case work have become an important aspect of policing, criminal justice, and victim identification. In multiple ways, forensic genetics contributes to rendering these domains more agile vis-à-vis existing and emerging demands and challenges. Novel technologies are developed in response to scientific insights and practical problems, while remaining subject to the same responsibilities that scientists and users have towards their specific institutions, disciplines, and to society as a whole. Therefore, in this paper, we offer an approach through which the forensic genetics community can reflect on their daily decisions in both scientific research and case work as part of their professional and ethical forensic practice. While we cannot, nor do we aim to, provide technical or legal solutions, we offer a framework for ethical conduct.”
< Indeed, it would be interesting to compare the ethical risk of the application of these technologies and the actual benefit that these technologies bring to the resolution of criminal cases.>
Thank you for this comment. We address the strong position in policing, criminal justice and society that these technologies hold, and what they can contribute to investigations by examples. In the context of the rich existing scientific literature on the advantages of forensic genetics and its developments, we feel that we do not need to further elaborate on that point.
However, to clarify our thinking, we have added several paragraphs throughout, including in section 1, p. 2:
“​​​​In some parts of the world, forensic genetics research and case work have become an important aspect of policing, criminal justice, and victim identification. In multiple ways, forensic genetics contributes to rendering these domains more agile vis-à-vis existing and emerging demands and challenges. Novel technologies are developed in response to scientific insights and practical problems, while remaining subject to the same responsibilities that scientists and users have towards their specific institutions, disciplines, and to society as a whole.”
We have added a paragraph at the beginning of the section on FDP and BGA to clarify some of our thinking here (section 7.1., page 10).
“FDP and BGA have provided intelligence to criminal investigations that, at times, has resulted in the detection of suspects. To date, in the USA, hundreds of investigations have drawn on intelligence from FDP and BGA analyses since 2015. In Europe, these techniques (especially BGA), have found application in cases since at least 1999, including in the Marianne Vaatstra case (M’charek et al. 2008); the investigation of the Madrid train bombings of 2004 (cf. Phillips et al. 2009); the Phantom of Heilbronn investigation in 2008 [71]; and the detection of the murder of Eva Blanco Puig in Spain in 2015. In the Vaatstra case in 1999, an early form of BGA was used to the effect of taking public pressure off asylum seekers who had been considered prime suspects. In the Madrid investigation, the BGA analysis contributed some intelligence to the investigation. In the case of the Phantom of Heilbronn, the results did not contribute to the detection of perpetrators; rather, it seemed to fuel the bias of investigators. While the case of the Phantom was widely debated, unsuccessful applications of FDP or BGA tend not to be discussed publicly. Yet insight into limitations can assist the wider forensic genetics community of scientists and users, but also publics, in understanding the effective parameters of these techniques.”
In the section on FGG we also outline the several cases that have been solved through this technology (section 7.2, page 12):
“... several cases have now been solved through FGG internationally. According to Kling et al. (2021), FGG has been used to generate investigative leads in nearly 200 cold cases and some active investigations (see also Greytak et al. 2019, Katsanis 2020, Kennett 2019). In Sweden, prosecutors have allowed investigators to use consumer genealogy databases to solve cold cases: missing persons and a criminal case [78]. In Canada two cold cases have been solved through forensic genealogy [8]. In the UK, a study has assessed the likely effectiveness of long-range familial searches through GEDmatch [79]. The Golden State Killer also sparked interest in this procedure among citizens and the companies offering these services, with existing databases now allowing for and, in some cases, encouraging law enforcement searches, as in the case with GEDmatch (recently acquired by commercial forensic service provider Verogen which raises further ethical issues which we do not have the space to explore here) and FamilyTreeDNA. In addition, services exclusively dedicated to law enforcement searches, e.g. DNASolves, DNA.Land, Geneanet, and Geni, are emerging [80] and some law enforcement agencies are setting up their own FGG services.”
<Authors should insert in the manuscript the advantages that forensic genetics provides to society.>
We have addressed this point above.
<Several concepts in the paper are redundant.>
We appreciate the comment but are puzzled by it. In our view, all concepts used are essential to this paper, they are vital in order to show our approach and provide evidence. Without further elaboration on which concepts are, in the reviewer’s view, redundant, we are not able to address this comment.
<The references in the paper should be reviewed.>
Thank you for the suggestion. We have reviewed our references and added several references as indicated in the sections above, and have also followed the recommendation by Reviewer 2 to include new references (please see reviewer 2 report).
Reviewer 2 Report
This article is of high value in the field of forensic genetics. The issues raised are very relevant and timely. I have only minor suggested edits:
- Page 2, para 3: while the focus of the article is ethics as an instrict part of forensic science, it is important to note that some of the capabilities discussed (particularly FGG) do not entirely sit in the forensic domain. This is an even greater risk, and should perhaps be called out.
- Page 3, last para - While I agree with the statement about data protection being an inadequate proxy for privacy, this statement is made without context or rationale. I would suggest softening this language, referencing a source here, or providing a more comprehensive of why this is the case (the last option would both lengthen and divert attention from the key argument of the paper, so would not be recommended).
- Page 8 - Para 1 moves from discussion of intelligence-related uses to presentation in court. I think there needs to be a lead-in to explain how these capabilities might end up in evidence, when STR-based testing may subsequently be undertaken. Situations like use of FDP or FGG in applications for warrants could be considered.
- Page 8 - Para 2 - apostrophe missing after 'correct
- Page 8 - Para 2 - discussion around arbitrary or inconsistent exclusion of evidence should include a reference. It reads as a statement without a source, particularly claiming this "often" occurs.
- Page 11 - Para 2 - 'social media and other records'. Given FGG is highly dependent on official and genealogical records, supported by social media and similar sources, I would suggest rephrasing to highlight the primary source would be online genealogy database records (which, of itself, raises ethical issues).
- Page 11 - Para 2 - the statement that 'several' cases have been solved, and citing Katsanis, appears out of date in a 2021 paper. Consider some of the more recent references (see Kling, D., et al. (2021). "Investigative genetic genealogy: Current methods, knowledge and practice." Forensic Science International: Genetics: 102474, references 15-18).
- Page 11 - Para 2 - 4th last line - Family Tree DNA should be FamilyTreeDNA.
- Page 11 - last para and Page 12, para 1 - this part of the article reads as though users of genealogy platforms have no option but to make their data available for law enforcement matching. While consent here is often unclear, and inconsistent, it should at least be noted that most sites have considered this issue and do now seek some form of consent to match with law enforcement profiles.
Author Response
Dear Professor Morling, dear Ms. Shen, dear reviewer,
Many thanks to you for your time reading and commenting upon our manuscript. We appreciate the opportunity to revise. We have made revisions in line with suggestions, and present our responses here in the format <reviewer’s comment> / our response. Here, we include new references in Harvard style for ease of identification, but in the manuscript we apply the journal’s endnote approach. We include all material responses as tracked changes in the manuscript upload. We look forward to hearing from you.
<This article is of high value in the field of forensic genetics. The issues raised are very relevant and timely.>
Many thanks for your supportive comment.
<Page 2, para 3: while the focus of the article is ethics as an intrinsic part of forensic science, it is important to note that some of the capabilities discussed (particularly FGG) do not entirely sit in the forensic domain. This is an even greater risk, and should perhaps be called out.>
We thank the reviewer for bringing this to our attention. We have revised the text accordingly (section 7.2, page 12):
“Investigators found partial DNA matches assumed to belong to distant relatives, which enabled the creation of a ‘family tree’, primarily based on online genealogy database records, and including information from sources such as social media and other records. The data of these records sit outside of the forensic domain, and their use in policing raises a number of legal and ethical questions, including on the level of appropriate consent; dual/unintended use of data; and the whether the capacity exists or can be introduced for blocking or deleting data from such sources if data subjects disagree with law enforcement uses of their data.”
<Page 3, last para - While I agree with the statement about data protection being an inadequate proxy for privacy, this statement is made without context or rationale. I would suggest softening this language, referencing a source here, or providing a more comprehensive of why this is the case (the last option would both lengthen and divert attention from the key argument of the paper, so would not be recommended).>
We appreciate the recommendation, and have, both, softened the language and included two references here (section 3, page 4):
“Secondly, within this “thin” understanding of ethics, ‘privacy’ tends to be perceived as an interpretable legal concept, addressed by compliance with law. Moreover, ‘data protection’ is often used as a proxy for ‘privacy’, whereby observance with principles of data protection will suffice, yet ‘privacy’ is far more expansive and manifold than mere data protection (Kahn 2003, Presidential Commission for the Study of Bioethical Issues 2012, pp. 33-50).
<Page 8 - Para 1 moves from discussion of intelligence-related uses to presentation in court. I think there needs to be a lead-in to explain how these capabilities might end up in evidence, when STR-based testing may subsequently be undertaken. Situations like use of FDP or FGG in applications for warrants could be considered.>
We are grateful to the reviewer for identifying this issue, and we have addressed this in more detail (section 6., page 8):
“Assuming that DNA science moves from investigation onto a later life stage: forensic intelligence from genetic genealogy, but perhaps also from other techniques, may lead to requests by police for arrest warrants. This potential outcome would require of such forensic intelligence to be carefully articulated, and to take into account the context in which it may be used, including institutional policing cultures and wider political debates, e.g. about migration and criminality. In inquisitorial systems (but highly unlikely ever in adversarial systems), forensic intelligence may even make it into the report submitted to the investigating judge. At some point, if rarely, forensic experts may be called as witness to explain the intelligence.
We know for forensic evidence that the judge may be required to take on a “gatekeeper” role to ensure that “junk science” remains inadmissible at trial. This role and its fulfilment has been highly criticized already for adversarial systems where mechanism are in place for contestation [56], [57], with conclusions drawn that “scientific illiteracy on the part of the legal profession, when coupled with the flaws in forensic science, forms a ‘toxic combination’” [56, p. 82]. The trial process rarely affords even the scientifically literate much opportunity to interpret and apply ‘correctly’ complex tests (assuming there is a ‘correct’ application) for whether scientific evidence is ‘sufficiently reliable’ (Edmond 2020). In addition, such admissibility testing may come too late in the criminal process. Consider, too, that in the inquisitorial system forensic expertise is very rarely challenged. Even where challenges to scientific information exist, decisions to admit such information at trial can appear arbitrary, or inconsistent, with decisions based upon non-scientific criteria (Edmond et al. 2016).”
<Page 8 - Para 2 - apostrophe missing after 'correct>
Thank you, this oversight has been corrected.
<Page 8 - Para 2 - discussion around arbitrary or inconsistent exclusion of evidence should include a reference. It reads as a statement without a source, particularly claiming this "often" occurs.>
Thanks for this suggestion. We have made the requested changes (section 6. page 8):
“Even where challenges to scientific information exist, decisions to admit such information at trial can appear arbitrary, or inconsistent, with decisions based upon non-scientific criteria (Edmond et al. 2016).”
<Page 11 - Para 2 - 'social media and other records'. Given FGG is highly dependent on official and genealogical records, supported by social media and similar sources, I would suggest rephrasing to highlight the primary source would be online genealogy database records (which, of itself, raises ethical issues).>
Many thanks for pointing this out. We have revised the sentence (section 7.2, page 12):
“Investigators found partial DNA matches assumed to belong to distant relatives, which enabled the creation of a ‘family tree’, primarily based on online genealogy database records, and including information from sources such as social media and other records. The data of these records sit outside of the forensic domain, and their use in policing raises a number of legal and ethical questions, including on the level of appropriate consent; dual/unintended use of data; and whether the capacity exists or can be introduced for blocking or deleting data from such sources if data subjects disagree with law enforcement uses of their data.”
<Page 11 - Para 2 - the statement that 'several' cases have been solved, and citing Katsanis, appears out of date in a 2021 paper. Consider some of the more recent references (see Kling, D., et al. (2021). "Investigative genetic genealogy: Current methods, knowledge and practice." Forensic Science International: Genetics: 102474, references 15-18).>
Thank you for this recommendation, we have added a number of references (section 7.2, page 12):
“... several cases have now been solved through FGG internationally. According to Kling et al. (2021), FGG has been used to generate investigative leads in nearly 200 cold cases and some active investigations (see also Greytak et al. 2019, Katsanis 2020, Kennett 2019). In Sweden, prosecutors have allowed investigators to use consumer genealogy databases to solve cold cases: missing persons and a criminal case [78]. In Canada two cold cases have been solved through forensic genealogy [8]. In the UK, a study has assessed the likely effectiveness of long-range familial searches through GEDmatch [79]. The Golden State Killer also sparked interest in this procedure among citizens and the companies offering these services, with existing databases now allowing for and, in some cases, encouraging law enforcement searches, as in the case with GEDmatch (recently acquired by commercial forensic service provider Verogen which raises further ethical issues which we do not have the space to explore here) and FamilyTreeDNA. In addition, services exclusively dedicated to law enforcement searches, e.g. DNASolves, DNA.Land, Geneanet, and Geni, are emerging [80] and some law enforcement agencies are setting up their own FGG services.”
<Page 11 - Para 2 - 4th last line - Family Tree DNA should be FamilyTreeDNA.>
Thank you, this oversight has been corrected.
<Page 11 - last para and Page 12, para 1 - this part of the article reads as though users of genealogy platforms have no option but to make their data available for law enforcement matching. While consent here is often unclear, and inconsistent, it should at least be noted that most sites have considered this issue and do now seek some form of consent to match with law enforcement profiles.>
Thank you for pointing this out. We have clarified this (Section 7.2, page 13):
“However, it should be noted that different databases have varying policies in terms of making genetic data available for law enforcement searches (Skeva et al. 2020). GEDmatch and FamilyTreeDNA encourage such searches. GEDmatch offers users to opt-in while FamilyTreeDNA, in 2019, instituted an opt-out policy that automatically opens up users’ data to law enforcement searches. However, under jurisdiction of the European Union’s General Data Protection Regulation (GDPR), users from current and recent European Union member states who have profiles on FamilyTreeDNA have been automatically opted out. This means that European Union citizens must opt-in if they wish to have their DNA profile included for criminal investigation purposes (Granja 2020, UK Biometrics and Forensic Ethics Group 2020). Adopting a different stance, companies such as AncestryDNA and 23andMe have categorically denied access for law enforcement agents to such data, unless subpoenaed by a court order to do so (Skeva et al. 2020).
Round 2
Reviewer 1 Report
Page 6. Is the position of the "14" reference in section 5 correct?